# CD8A as a Prognostic and Immunotherapy Predictive Biomarker Can Be Evaluated by MRI Radiomics Features in Bladder Cancer

**DOI:** 10.3390/cancers14194866

**Published:** 2022-10-05

**Authors:** Zongtai Zheng, Yadong Guo, Xiongsheng Huang, Ji Liu, Ruiliang Wang, Xiaofu Qiu, Shenghua Liu

**Affiliations:** 1Department of Urology, Guangdong Second Provincial General Hospital, Guangzhou 510317, China; 2Urologic Cancer Institute, Tongji University School of Medicine, Shanghai 200072, China; 3Department of Urology, Shanghai Tenth People’s Hospital, School of Medicine, Tongji University, Shanghai 200072, China; 4First Clinical College, Guangdong Medical University, Zhanjiang 524023, China

**Keywords:** bladder cancer, CD8A, MRI, radiogenomics, immunotherapy, prognosis

## Abstract

**Simple Summary:**

It is necessary to explore a reliable predictive biomarker for immunotherapeutic strategies in bladder cancer. As an important member of tumor-infiltrating immune cells, cytotoxic T cells are the ultimate effectors of cancer rejection because of their specificity against cancer cells and cytolytic capabilities. Based on RNA expression data, we found that among T cytotoxic pathway-related genes, CD8A is a novel protective gene in bladder cancer. CD8A expression is highly associated with tumor microenvironment characteristics and tumor mutation burden, which partially explain CD8A as a prognostic and immunotherapy predictive biomarker in bladder cancer. In addition, we performed the radiogenomics in bladder cancer based on the RNA-sequence data and MRI-based radiomics features in our center. In this way, an MRI-based radiomics signature has the potential to preoperatively predict the expression of CD8A, thereby contributing to the preoperative prediction of prognosis and immunotherapeutic susceptibility in bladder cancer.

**Abstract:**

As an important member of T cytotoxic pathway-related genes, CD8a molecule (CD8A) may be a useful biomarker of immunotherapeutic response and immune cell infiltration. We aimed to investigate the clinical predictive value of CD8A in prognosis and tumor microenvironment (TME) and preoperatively predict the expression of CD8A using radiogenomics in bladder cancer (BCa). Among 12 T cytotoxic pathway-related genes, CD8A was a novel protective gene and had the highest correlations with T cells and Macrophages M1 in BCa. In advanced cancer patients treated with immunotherapy, low CD8A expression was associated with immunotherapeutic failure and poor survival outcomes. CD8A expression was highly related to tumor mutation burden, critical immune checkpoint genes and several types of tumor-infiltrating immune cells, predicting effective response to immunotherapy. The preoperative MRI radiomics features and RNA-sequence data of 111 BCa samples were used to develop a radiomics signature that achieved good performance in the prediction of CD8A expression in both the training (area under curve (AUC): 0.857) and validation sets (AUC: 0.844). CD8A is a novel indicator for predicting the prognosis and immunotherapeutic response in BCa. A radiomics signature has the potential to preoperatively predict the expression of CD8A in BCa patients.

## 1. Introduction

Bladder cancer (BCa) is one of the most malign neoplasia in the world, with high morbidity and mortality [1]. Approximately 30% of BCa patients have muscle-invasive bladder carcinoma (MIBC) at the time of diagnosis with a high rate of metastasis and mortality [2,3], and the other non-muscle-invasive bladder carcinoma (NMIBC) patients are characterized by a high rate of recurrence and a certain risk of progression to MIBC [4]. Treatment strategies differ greatly between NMIBC and MIBC. NMIBC patients are treated with transurethral resection followed by intravesical therapy with Bacillus Calmette–Guérin (BCG) or intravesical chemotherapy for carcinoma in situ or high-grade T1 [5], while MIBC patients (>T1) are treated with cisplatin-based neoadjuvant chemotherapy and radical cystectomy [6]. Despite the development in adjuvant therapy and surgical techniques, some BCa patients do not benefit from the recommended treatment strategies, with only an 8% 5-year survival rate in BCa patients with metastasis [7]. Therefore, it is urgently needed to investigate new and reliable therapeutic approaches for BCa patients.

Immune checkpoint inhibitor (ICI) is becoming a useful and important immunotherapy in BCa [8]. It is reported that Nivolumab (programmed cell death protein 1 (PD-1) inhibitor) could induce neoplastic cell death in urothelial bladder cancers with metastases [9]. In addition, immunotherapy had been reported to be a useful treatment for both early and late stages of BCa [10,11]. However, the results of clinical trials revealed that only a small fraction of BCa patients benefit from immunotherapy [8,12,13]. The role of inhibitory receptors expression such as programmed death ligand-1 (PD-L1) as a predictor of immunotherapeutic response had not been determined [8]. Thus, it is necessary to explore useful immunotherapy targets to identify subgroups of BCa patients that benefit from immunotherapy.

As members of tumor-infiltrating immune cells (TIICs), cytotoxic T cells (CD8+ T cells) are recruited to the tumor microenvironment (TME) to play an important role in tumor adaptive immunity during tumor immune responses. Cytotoxic T cells are the ultimate effectors of cancer rejection because of their specificity against cancer cells and cytolytic capabilities [14]. The CD8a molecule (CD8A) is a member of T cytotoxic pathway-related genes and encodes the CD8 antigen that is a cell surface glycoprotein found on most cytotoxic T cells. The CD8 antigen acts as a coreceptor with the T-cell receptor on the T cell to recognize antigens displayed by an antigen-presenting cell in the context of class I MHC molecules. CD8A expression may be a useful and measurable predictive marker of immunotherapeutic response and immune cell infiltration [15]. A previous study for pan-cancer has reported that high status of CD8A with high expression of PD-L1 might be a predictive marker of immunotherapeutic response [16]. However, the prognostic value of CD8A and the association between CD8A expression and immunotherapeutic response have not been explored in BCa.

Magnetic resonance imaging (MRI) is emerging as a routine examination for BCa patients in clinical practice. It can intuitively demonstrate the size, number, location and shape of lesions. The Vesical Imaging Reporting and Data System (VI-RADS) is a five-point assessment scale based on multiparametric MRI for describing the likelihood of muscle invasion [17]. It has been consistently validated across multiple institutions as a useful tool for local staging in BCa. However, the VI-RADS is generated by the radiologist’s subjective evaluation, which limits its application in predicting the biological and molecular characteristics of tumors. In contrast, an emerging technique termed radiogenomics may be a useful tool in clinical oncology. It can extract high-throughput quantitative imaging features not perceptible to the human eye from digital medical images and provide an objective and non-invasive approach in uncovering the TME characteristics in a tumor. A previous study developed a CT-based radiomics signature to predict the CD8 cell’s tumor infiltration and immunotherapeutic response in patients with advanced solid tumors [18]. Preoperative evaluation of TME characteristics using radiogenomics may be a useful way to facilitate survival prediction and personalized immunotherapy. At present, there are no studies using radiogenomics to predict the CD8A expression in BCa.

Therefore, the aim of this study was to assess the clinical predictive value of CD8A in prognosis and immunotherapeutic response, investigate the association between CD8A and TME and construct an MRI-based radiomics signature for the prediction of CD8A expression using radiogenomics in BCa.

## 2. Materials and Methods

### 2.1. Data Acquisition

Figure 1 shows a work flow of the present study.

For BCa datasets from the Gene Expression Omnibus (GEO, https://www.ncbi.nlm.nih.gov/, accessed on 6 April 2022) and The Cancer Genome Atlas (TCGA, http://cancergenome.nih.gov/, accessed on 10 April 2022), patients who met the following inclusion criteria were selected: (1) histologically diagnosed with BCa; (2) available RNA expression data; (3) available prognostic information. In this study, 405 patients from TCGA, 165 patients from GSE13507, 93 patients from GSE31684, 73 patients from GSE48075, 224 patients from GSE32894 and 73 patients from GSE48277 were included. In addition, two datasets (IMvigor210C and GSE93157) with advanced cancer patients treated with immunotherapy were also used to investigate the potential clinical value of CD8A in classifying the immunotherapy susceptibility for patients. IMvigor210 dataset consisted of patients with metastatic urothelial cancer treated with atezolizumab (PD-L1 inhibitor) [19], and the RNA expression data and prognostic information were downloaded from the website http://research-pub.gene.com/IMvigor210CoreBiologies, accessed on 10 April 2022. GSE93157 dataset included 65 patients with melanoma, lung cancer, and head and neck cancer treated anti-PD1 [20]. Appendix A summarizes the detailed information on these datasets.

In Shanghai Tenth People’s Hospital, BCa patients from November 2019 to July 2021 were collected to develop MRI-based radiomics signature for the prediction of CD8A expression. The inclusion criteria in our center included the following: (1) histologically diagnosed with BCa; (2) available RNA-sequence data; (3) available MRI within 20 days before surgery. The exclusion criteria were as followed: (1) poor-quality MRI images; (2) any treatments were performed before MRI examination; (3) tumors for which it was difficult to define the boundaries; (4) lack of baseline clinical factors. The protocols of MRI acquisition, total RNA extraction, paired-end libraries generation and RNA-sequence can be obtained in our previous methods [21,22].

The “BIOCARTA_T CYTOTOXIC_PATHWAY” gene list including 12 T cytotoxic pathway-related genes were obtained from the Gene Set Enrichment Analysis (GSEA) database (https://www.gsea-msigdb.org/gsea/msigdb/cards/BIOCARTA_TCYTOTOXIC_PATHWAY, accessed on 10 April 2022).

### 2.2. HEmatoxylin and Eosin (H&E) and Immunohistochemistry (IHC) Staining

After fixation, BCa tumor tissues were embedded in paraffin using standardized cassettes, sectioned at 5 μm, and stained with H&E and IHC as previously described in detail [23,24]. Rabbit anti-human monoclonal primary antibodies against CD8A (CST, cat# 85336) were utilized to detect CD8A expression according to the manufacturer’s protocol.

### 2.3. Evaluation of TIICs and TME

CIBERSORT analysis was performed to evaluate the relative proportion of 22 TIICs based on the RNA expression data of marker genes in each tumor sample (https://cibersort.stanford.edu, accessed on 10 April 2022). The TCGA dataset that has the maximum number of BCa patients was used to investigate the different TIICs between low and high CD8A expression groups. The relative proportion of 22 TIICs was also evaluated in our center based on the RNA-sequence data. The TME scores (including immune score, stromal score and estimate score) of each BCa patient were calculated using the R package “ESTIMATE”.

### 2.4. Association of CD8A with TIICs and Critical Immune Checkpoint Genes

We used the tumor immune estimation resource (TIMER, https://cistrome.shinyapps.io/timer/, accessed on 10 April 2022) to assess the correlation between CD8A and some TIICs including B cells, CD8+ T cells, CD4+ T cells, macrophages, neutrophil and dendritic cells. In addition, the correlation between CD8A and critical immune checkpoint genes including PDCD1, CTLA4, LAG3, TIGIT, HAVCR2, CD274 and TNFRSF9 was also investigated using TIMER.

### 2.5. Association of CD8A with Tumor Mutation Burden (TMB), Stemness and Microsatellite Instability (MSI)

The UCSCXenaShiny (https://hiplot.com.cn/advance/ucsc-xena-shiny, accessed on 10 April 2022) was used to explore the association of CD8A with TMB, stemness and MSI among 32 cancer types. In addition, the scatter plot showing the correlation between CD8A expression and TMB in TCGA-BCa was also performed.

### 2.6. Association between CD8A and Drug Sensitivity

The drug sensitivity of each BCa sample was predicted based on the Genomics of Drug Sensitivity in Cancer, which is the largest public pharmacogenomics database. Seven drugs for anti-cancer treatment were selected to evaluate the IC50 of each sample using the “pRRophetic” R package.

### 2.7. Survival Analysis and Meta-Analysis

The prognostic value of CD8A expression among 33 types of tumors in TCGA was analyzed using UCSCXenaShiny (https://hiplot.com.cn/advance/ucsc-xena-shiny, accessed on 10 April 2022). The Kaplan–Meier and log-rank tests were used to evaluate the prognostic value of CD8A expression in TCGA, IMvigor210 and six GEO datasets (GSE93157, GSE13507, GSE31684, GSE48075, GSE32894 and GSE48277). Patients were divided into low and high CD8A expression groups based on the optimal cutoff values in each dataset calculated by the “survminer” R package v0.4.8 (Marseille, France). Univariate and multivariate Cox regression analyses were used to obtain the independent prognostic factor in TCGA. In addition, the results of univariate Cox regression analyses (including HR and 95% confidence interval (CI)) in seven BCa datasets (TCGA, IMvigor210, GSE13507, GSE31684, GSE48075, GSE32894 and GSE48277) were used for meta-analysis. I^2^ statistic and χ^2^-based Q test were used to evaluated the heterogeneity of meta-analysis. If I^2^ value > 25%, the meta-analysis was regarded as high heterogeneity and the random-effect model was used for meta-analysis. The funnel plot, Begg’s test and Egger’s test were applied to investigate the possible publication bias. A *p* value > 0.05 was regarded as no publication bias.

### 2.8. Construction of the Nomogram

Factors that were significant in multivariate Cox analysis were used to construct a predictive nomogram. Time-dependent receiver operating characteristic (ROC) curves of the nomogram were used to evaluate its performance in survival prediction. Decision curve analysis (DCA) and calibration plots were performed to investigate the clinical utility and performance of the nomogram, respectively.

### 2.9. GSEA and Gene Set Variation Analysis (GSVA)

GSEA (http://www.broadinstitute.org/gsea/index.jsp, accessed on 10 April 2022) (Los Angeles, CA, USA) and GSVA (R package “GSVA” and “GSEABase”) were performed to investigate different biological pathways between different groups. A nominal *p* < 0.05 and a false discovery rate < 0.25 were regarded as significant.

### 2.10. Region of Interest (ROI) Segmentation and Radiomics Feature Extraction

A radiologist (F Xu) with over 5 years of experience in bladder MRI reading used the ITK-SNAP software (version 3.6.0; http://itk-snap.org, accessed on 10 April 2022) (Kitware Inc., Clifton Park, NY, USA) to manually outline the ROIs of lesions layer by layer on T2WI images and the delay phase of DCE images. If a BCa patient had multiple lesions, the largest lesion was uniquely outlined for feature extraction. After 30 days, the lesions of 40 randomly selected patients were repeatedly delineated by the same radiologist (F Xu) and another radiologist (T Xu) to evaluate the interobserver reliability.

Radiomics features were extracted from the ROI of each BCa using the Python package Pyradiomics v.2.2.0 (http://www.radiomics.io/pyradiomics.html, accessed on 10 April 2022) (Boston, MA, USA). For each MRI sequence, 18 first-order features, 14 morphology features, 14 gray level dependence matrix (GLDM) features, 24 gray level co-occurrence matrix (GLCM) features, 16 gray level size zone matrix (GLSZM) features and 16 gray level run length matrix (GLRLM) features were extracted on the base images; 25 NGTDM features, 80 GLSZM features, 90 first-order features, 120 GLCM features, 80 GLRLM features and 70 GLDM features were extracted on the Laplacian of Gaussian (LoG)-filtered images with various sigma values (0.5, 1.0, 2.0, 3.0 and 4.0) with 25 bins; 40 NGTDM features, 112 GLDM features, 144 first-order features, 192 GLCM features, 128 GLRLM features and 128 GLSZM features were extracted on the wavelet-filtered images; 5 NGTDM features, 14 GLDM features, 24 GLCM features, 144 first-order features, 16 GLRLM features, 16 GLSZM features and 5 NGTDM features were calculated on the exponential, logarithm, square, square root and gradient images, respectively. In total, 1781 radiomics features were extracted from DCE and T2WI images, respectively. Each radiomics feature was normalized with a Z-score before further analyses.

### 2.11. Feature Selection and Radiomics Signature Construction

BCa patients in Shanghai Tenth People’s Hospital were divided into low and high CD8A groups based on the median value of CD8A expression. Then, these patients were randomly allocated into training set and validation set based on a 7:3 ratio.

The intra- and interclass correlation coefficients (ICCs) were calculated between two radiologists to assess the interobserver reliability of radiomics features. The minimum redundancy maximum relevance (mRMR) algorithm was performed to select the relevant and non-redundant features and rank the importance of features [25]. In this way, the top 10 features were obtained to develop radiomics signature for predicting the CD8A expression.

In this study, we used the least absolute shrinkage and selection operator (LASSO) algorithm to construct radiomics signature. The LASSO algorithm can remove redundant features and select features with non-zero coefficients for model construction. The radiomics score was then calculated for each BCa patient by summing the values of radiomics features weighted by their respective coefficients. The radiomics signature was developed via ten cross-validations in the training set to select the optimal values of λ and was assessed in the validation set. The area under the ROC curve (AUC), accuracy, sensitivity, specificity, negative predictive value (NPV) and positive predictive value (PPV) were used to assess the performance of the radiomics signature.

### 2.12. Statistical Analysis

The *t*-test, Wilcoxon test or one-way ANOVA was properly performed to assess association between different factors. Meta-analysis was performed using the “metafor” R package v2.4 and “meta” R package v4.16. The “corrplot” R package v0.84 was used to generate correlation matrix. The “glmnet” R package v4.1 was used to develop the LASSO model. The “fmsb” R package v0.7 (Kobe, Japan) was used to plot the comprehensive performance of the LASSO model. SPSS 23.0 (SPSS, Armonk, NY, USA) and R v3.6.1 (https://www.r-project.org/, accessed on 10 April 2022) (R Statistical Foundation, Vienna, Austria) were employed to conduct statistical analyses. A two-sided *p* value < 0.05 was considered significant.

## 3. Results

### 3.1. Analysis of the T Cytotoxic Pathway-Related Genes in BCa

Figure 2A demonstrates the locations of the T cytotoxic pathway-related genes on their respective chromosomes. Figure 2B,C shows the PPI network and correlation network among T cytotoxic pathway-related genes, which suggest that the expressions and functions of T cytotoxic pathway-related genes were highly correlated with each other. Among 12 T cytotoxic pathway-related genes, CD8A had the highest correlations with several TIICs, including T-cell CD8, T-cell CD4 naive, T-cell CD4 memory activated and Macrophages M1 (Figure 2D,E). Univariate Cox analysis of OS and DFS showed that CD8A was a novel prognostic factor among 12 T cytotoxic pathway-related genes (Figure 2F,G). In this way, CD8A was selected for further analysis. Patients with low CD8A expression were prone to have shorter overall survival (OS), disease-specific survival (DSS) and progression-free interval (PFI) than patients with high CD8A expression in this cohort of 33 cancer types (Appendix A). Then, we divided 8040 cancer patients with different types of cancers in TCGA into low CD8A expression and high CD8A expression groups based on the median value of CD8A gene expression and investigated the prognostic value of CD8A through the Kaplan–Meier method. The result showed that patients with low CD8A expression had significantly shorter OS than patients with high CD8A expression (Appendix A). In TCGA-BCa, patients with low CD8A expression had significantly shorter DFS than patients with high CD8A expression (Appendix A).

Pan-cancer analysis showed that CD8A was highly associated with TMB, stemness and MSI among several tumor types including BCa (Appendix A). We further investigated the association between CD8A expression and IC50 values of seven drugs for anti-cancer treatment. The results showed that the IC50 values of Docetaxel, Camptothecin, Paclitaxel, Gemcitabine, FGFR inhibitor, Cisplatin and Tamoxifen were significantly lower in patients with high CD8A expression (Appendix A), which revealed that patients with high CD8A expression were more sensitive to these drugs.

### 3.2. The Prognostic Value of CD8A

In three BCa datasets including TCGA, GSE48277 and GSE48075, BCa patients with low CD8A expression still had shorter OS than patients with high CD8A expression (Figure 3A–C). In TCGA, BCa patients in the low CD8A expression group had a significantly higher percentage of lymph node metastasis (*p* = 0.015, Appendix A) and death (*p* = 0.048, Appendix A). In two immunotherapy datasets including IMvigor210C and GSE93157, CD8A acted as a protective gene for cancer patients treated with immunotherapy (Figure 3D,E). In addition, patients in the high CD8A expression group had a significantly higher percentage of partial response (PR)/complete response (CR) than those in the low CD8A expression group in IMvigor210 (Figure 3F, 33.9% and 20.2%, respectively, *p* = 0.028). In addition, we performed meta-analysis and Cox regression analysis to comprehensively investigate the prognostic value of CD8A. The results of meta-analysis based on seven BCa datasets revealed that CD8A was a protective gene for BCa patients (Figure 3G, Appendix A) without publication bias (Appendix A, Begg’s test: *p* = 0.453, and Egger’s test: *p* = 0.890). Univariate and multivariate Cox regression analysis showed that CD8A was an independent prognostic factor in BCa (Figure 3H).

A CD8A-based nomogram was developed to predict the 1-, 3- and 5-year OS rates of BCa patients (Figure 4A). ROC curves revealed that the nomogram had a good performance in survival prediction with the 1-, 3- and 5-year AUC values of 0.679, 0.722 and 0.722, respectively (Figure 4B). Calibration curves revealed that the prognostic predictions of the CD8A-based nomogram were similar to the actual 1-, 3- and 5-year OS (Figure 4C–E). DCA demonstrated that the CD8A-based nomogram had the optimal clinical net benefit compared to N stage and T stage (Figure 4F–H).

### 3.3. The potential Biological Mechanisms of the CD8A

GSEA revealed that several immune-associated pathways, including NATURAL_KILLER_CELL_MEDIATED_CYTOTOXICITY, T_CELL_RECEPTOR_SIGNALING_PATHWAY, B_CELL_RECEPTOR_SIGNALING_PATHWAY, REGULATION_OF_LYMPHOCYTE_ACTIVATION, REGULATION_OF_IMMUNE_EFFECTOR_PROCESS, T_CELL_ACTIVATION and POSITIVE_REGULATION_OF_IMMUNE_EFFECTOR_PROCESS, were mainly enriched in the high CD8A expression group (Figure 5A). GSVA also showed similar results with the ANTIGEN_PROCESSING_AND_PRESENTATION, NATURAL_KILLER_CELL_MEDIATED_CYTOTOXICITY and T_CELL_RECEPTOR_SIGNALING_PATHWAY highly enriched in the high CD8A expression group (Figure 5B).

### 3.4. Association of CD8A with TME Score, TIICs and Critical Immune Checkpoint Genes

The TME scores of low and high CD8A expression groups were calculated and compared. The results revealed that the high CD8A expression group had higher stromal scores, immune scores and TIDE scores than the low CD8A expression group (Figure 5C), and CD8A expression was positively related to stromal scores, immune scores and TIDE scores (Appendix A). Correlation matrix demonstrated that CD8A was positively related to some TIICs (CD8+ T cell, CD4+ T cell, Macrophages M1) and critical immune checkpoint genes (PDCD1, CTLA4, LAG3, TIGIT, HAVCR2, CD274 and TNFRSF9) (Figure 5D). Scatter plots also showed that CD8A expression was correlated to purity, CD8+ T cells, CD4+ T cells, neutrophil and dendritic cells (Appendix A). In addition, we investigated the association between CD8A and some critical immune checkpoint genes, and the results showed that CD8A expression was positively associated with the expression of critical immune checkpoint genes (Appendix A). Representative immunostainings for high and low CD8A expression in BCa are shown in Figure 5E,F, respectively.

We then investigated the correlation between CD8A expression and 20 TIICs in a cohort of 30 cancer types. The results revealed that CD8A expression was highly correlated to most of TIICs in pan-cancer (Appendix A). Obviously, CD8A expression was positively correlated to the proportions of CD8+ T cells and Macrophages M1 in pan-cancer (Appendix A).

In BCa, we compared the proportions of 22 TIICs between low and high CD8A expression groups in TCGA (Figure 6A, Appendix A). There were 13 of 22 TIICs differently distributed between low and high CD8A expression groups. Specifically, the high CD8A expression group had relatively higher percentages of CD8+ T cells and Macrophages M1 than the low CD8A expression group (Figure 6B, Appendix A).

### 3.5. Construction and Performance of the Radiomics Signature

In this study, 111 BCa patients with RNA-sequence data and preoperative MRI were collected in our center. There were 77 and 34 BCa patients in the training and validation sets, respectively. Table 1 shows the clinical characteristics of BCa patients. There were no statistical differences between training and validation sets.

The ICCs of radiomics features were between 0.762 and 0.908, suggesting a good inter- and intra-observer reproducibility. We then selected the top ten radiomics features through the mRMR algorithm to construct the radiomics signature. In this way, a LASSO model with nine radiomics features was constructed (Figure 7A,B). Appendix A demonstrated the coefficients of these nine features. Correlation analyses showed that these nine radiomics features in the LASSO model were not associated with each other (Appendix A, mean absolute Spearman ρ = 0.158). The LASSO model achieved the AUCs of 0.857 and 0.844 in the training and validation sets, respectively (Figure 7C), and the accuracies of 77.9% and 85.3% in the training and validation sets, respectively (Figure 7D).

Patients in the CD8A high expression group had significantly lower radiomics scores than patients in the CD8A low expression group in both training and validation sets (Appendix A, both *p* < 0.001). The waterfall plot revealed that patients with high radiomics scores had a strong tendency for CD8A low expression in the combined training and validation sets, which suggested that the CD8A expression of BCa patients could be correctly predicted based on the cutoff value of the LASSO model (Figure 7E).

### 3.6. The Potential Biological Mechanisms of the Radiomics-Predicted CD8A Expression

GSEA revealed that several immune-associated pathways, including LYMPHOCYTE_CHEMOTAXIS, LYMPHOCYTE_MEDIATED_IMMUNITY, NATURAL_KILLER_CELL_ACTIVATION_INVOLVED_IN_IMMUNE_RESPONSE, POSITIVE_REGULATION_OF_CELL_KILLING, POSITIVE_REGULATION_OF_T_CELL_MEDIATED_CYTOTOXICITY and POSITIVE_REGULATION_OF_T_CELL_PROLIFERATION, were mainly enriched in the radiomics-predicted high CD8A expression group (Figure 8A). GSVA also showed similar results with the ANTIGEN_PROCESSING_AND_PRESENTATION, NATURAL_KILLER_CELL_MEDIATED_CYTOTOXICITY, T_CELL_RECEPTOR_SIGNALING_PATHWAY and B_CELL_RECEPTOR_SIGNALING_PATHWAY highly enriched in the radiomics-predicted high CD8A expression group (Figure 8B). The relative proportions of 22 TIICs based on the RNA-sequence data in our center were evaluated using CIBERSORT analysis (Appendix A). The radiomics score of the LASSO model was negatively related to CD8A expression and degree of infiltration of CD8+ T cells and Macrophages M1 (Figure 8C–E, Appendix A), and the radiomics-predicted high CD8A expression group had relatively higher percentages of CD8+ T cell and Macrophages M1 than the radiomics-predicted low CD8A expression group (Figure 8F, Appendix A). In addition, all of the T cytotoxic pathway-related genes were differentially expressed between radiomics-predicted low and high CD8A expression groups (Appendix A).

## 4. Discussion

Previous studies had revealed the important role of the TME in the prognosis and immunotherapeutic response of tumor patients [26,27]. As a crucial component of the TME, TIICs act as a trigger for determining the sensitivity of tumor cells to immune system attack and immunotherapeutic response [28,29]. It has been reported that there were 11 types of TIICs significantly associated with OS in BCa [30]. Among TIICs, cytotoxic T cells (CD8+ T cells) as tumor antigen-specific cytotoxic T cells have a crucial role in the cytolytic killing of tumor cells. The low degree of infiltration of CD8+ T cells at the invasive margin and the tumor center was related to a poor OS in MIBC patients that underwent radical cystectomy [31]. In addition, CD8+ T cells can enhance antitumor activity to inhibit tumor cells through class I MHC [32] and enhance immunotherapy by activating CD4+ T cells through class II MHC [33]. On the cell membrane of CD8+ T cells, there is a cell surface glycoprotein named CD8 antigen that acts as a coreceptor with the T-cell receptor on the T cell to recognize antigens displayed by an antigen-presenting cell in the context of class I MHC molecules. CD8A encodes CD8 antigen, so CD8A expression may be associated with survival outcomes, immune cell infiltration and immunotherapeutic response in cancer patients. Thus, we aimed to explore the clinical predictive value of CD8A in prognosis and immunotherapeutic response and investigate the association between CD8A and TME in BCa.

All the T cytotoxic pathway-related genes were comprehensively analyzed in BCa to select the hub gene. As a result, CD8A was a novel gene that had the highest correlation with several TIICs and had the optimal prognostic value in BCa. Pan-cancer analysis, meta-analysis and multivariate Cox regression analysis were performed to further comprehensively explore the prognostic value of CD8A in cancer patients, especially BCa patients. The results revealed that CD8A was a useful prognostic factor among several types of cancers including BCa, and patients with low CD8A expression were prone to having poor survival outcomes. Previous study also revealed the protective role of CD8A in the prognoses of hepatocellular carcinoma, metastatic melanoma and head and neck squamous cell carcinoma [34,35,36]. However, few studies clarified the association between CD8A and immunotherapeutic response [37], so we further investigated the prognostic value of CD8A in cancer patients treated with immunotherapy in two datasets (IMvigor210 and GSE93157). Our results showed that low CD8A expression was associated with poor survival outcomes among cancer patients treated with immunotherapy, and patients in the high CD8A expression group had a higher percentage of PR/CR than those in the low CD8A expression group. Thus, CD8A can not only be a useful prognostic factor in BCa patients but also a predictive marker of immunotherapeutic response in cancer patients treated with immunotherapy. In addition, we also found that patients with high CD8A expression were more sensitive to several drugs for anti-cancer treatment, which may broaden the application of CD8A in clinical practice.

It is known that immune cell infiltration and activation are associated with the prognosis and immunotherapeutic response in BCa, so we used the bioinformatics analysis to investigate the relationship between CD8A and TIICs. GSEA and GSVA revealed that CD8A played an important role in the positive regulation of immune-associated pathways. In addition, CD8A expression was related to most types of TIICs, especially CD8+ T cells and Macrophages M1. Patients with high CD8A expression are prone to having a higher degree of infiltration of CD8+ T cells and Macrophages M1. As members of TIICs, CD8+ T cells and Macrophages M1 play a crucial role in the enhancement of antitumor activity and immunotherapeutic response, contributing to these two types of TIICs as prognostic and immunotherapy biomarkers in BCa. Jansen et al. reported that BCa patients with a high density of CD8+ T cells had favorite survival outcomes and strong responses to immunotherapy [38]. Zeng et al. reported that Macrophages M1 infiltration is a useful predictive marker for immunotherapeutic response and immunophenotype determination in metastatic urothelial cancer, and patients with high TMB and a high degree of Macrophages M1 infiltration had the best prognoses [39]. Thus, the positive association of CD8A with CD8+ T cells and Macrophages M1 partially explained the CD8A as a prognostic factor and a predictive marker of immunotherapeutic response in BCa.

Previous studies have reported some biomarkers related to cancer immunotherapeutic response such as TMB and the expression of some critical immune checkpoint genes [27,40], so we further explored the association of CD8A with TMB and critical immune checkpoint genes. Pan-cancer analysis revealed that CD8A was related to TMB in several types of tumors. In BCa, patients with higher CD8A expressions were prone to having higher levels of TMB. In addition, CD8A expression was positively related to some critical immune checkpoint genes, including PDCD1, CTLA4, LAG3, TIGIT, HAVCR2, CD274 and TNFRSF9. Therefore, the novel association of CD8A with TIICS, TMB and critical immune checkpoint genes contributes to the crucial role in TME and immunotherapeutic response, which facilitates CD8A as a useful prognostic marker and a predictor of immunotherapeutic response in BCa.

Due to the crucial role of CD8A in predicting the survival outcomes and immunotherapeutic response, we further used the MRI-based radiogenomics to preoperatively predict the expression of CD8A in BCa. The newly developed VI-RADS standardizes the imaging and reporting of BCa on MRI. Many institutions have validated its reliability as an appropriate tool for local staging and determining detrusor muscle invasion in BCa. However, it highly relies on the evaluation of largely morphologic features by radiologists’ assessment, such as the size of the lesion, and the enhancement and signal characteristics in digital medical images, which are insufficient and subjective. In addition, it is highly difficult to distinguish the extensive digital characteristics of tumor cells in digital medical images by simple quantitative methods and visual assessment [41]. Radiomics can extract high-throughput quantitative radiomics features that include comprehensive information of tumor biology such as tumor invasive growth patterns, necrosis and neovascularity. This technique is highly sensitive in distinguishing subtle changes in tumor morphology pathophysiology and proteomics, which cannot be deciphered by the human eye. Radiomics had been applied to preoperatively predict the muscle-invasive status, tumor grade, prognosis and treatment response in BCa [42], revealing that the radiomics has potential in predicting the risk groups of BCa patients in training and validation sets, suggesting that some of the MRI-based radiomics features have the potential to reflect the biological behavior on the onset of a tumor. In this study, radiomics features extracted from T2WI and DCE images were used to develop a radiomics signature to preoperatively predict the CD8A expression of BCa patients in our center. A LASSO model based on nine radiomics features achieved good performance in predicting the CD8A expression of BCa patients in both training and validation sets, indicating that the MRI-based radiomics features were associated with the biology of BCa. Interestingly, in this LASSO model, the numbers of radiomics features from T2WI and DCE were four and five, indicating that T2WI-based radiomics features and DCE-based radiomics features have the same importance in the assessment of CD8A expression. In addition, four of nine radiomics features in the LASSO model were wavelet-filtered features, which indicates that the wavelet transform filter has the novel ability to reflect genetic information in BCa. The wavelet transform filter can create eight decompositions per level in each of the three dimensions and generate high-dimensional wavelet-filtered features that cannot be deciphered by visual assessment. Compared with visual assessment by radiologists or low-level radiomics features, wavelet-filtered features include comprehensive information of tumor heterogeneity and tumor biology in several types of tumors, including prostate carcinoma [43], renal cell carcinoma [44], intrahepatic cholangiocarcinoma [45] and BCa [46].

We further investigated the biological behavior of the LASSO model based on the RNA-sequence data in our center. Interestingly, the radiomics score of the LASSO model was negatively related to CD8A expression and the degree of infiltration of CD8+ T cells and Macrophages M1. GSEA and GSVA further showed that several positive regulations of immune-associated pathways were highly enriched in the radiomics-predicted CD8A high expression group. These results revealed distinct TME between radiomics-predicted low and high CD8A expression groups and validated good performance of the LASSO model from an immunological perspective.

This study had limitations. One is that the mechanisms related to the regulation of TME by CD8A in BCa were not investigated, which will require additional mechanistic analyses in the future. Another limitation is that the number of BCa samples in our center is small. A larger number of BCa patients with preoperative MRI and RNA sequence should be collected to further assess the performance of the LASSO model in predicting CD8A expression.

## 5. Conclusions

In conclusion, our study illustrated that CD8A can be a useful indicator for predicting the survival outcomes and immunotherapeutic response in BCa. The expression of CD8A was associated with TME characteristics and TMB, which facilitates CD8A as a predictor of immunotherapeutic response. A radiomics signature based on nine MRI-based radiomics features has the potential to preoperatively predict the expression of CD8A in BCa patients, thereby contributing to the prediction of prognosis and immunotherapeutic susceptibility.

## Figures and Tables

**Figure 1 cancers-14-04866-f001:**
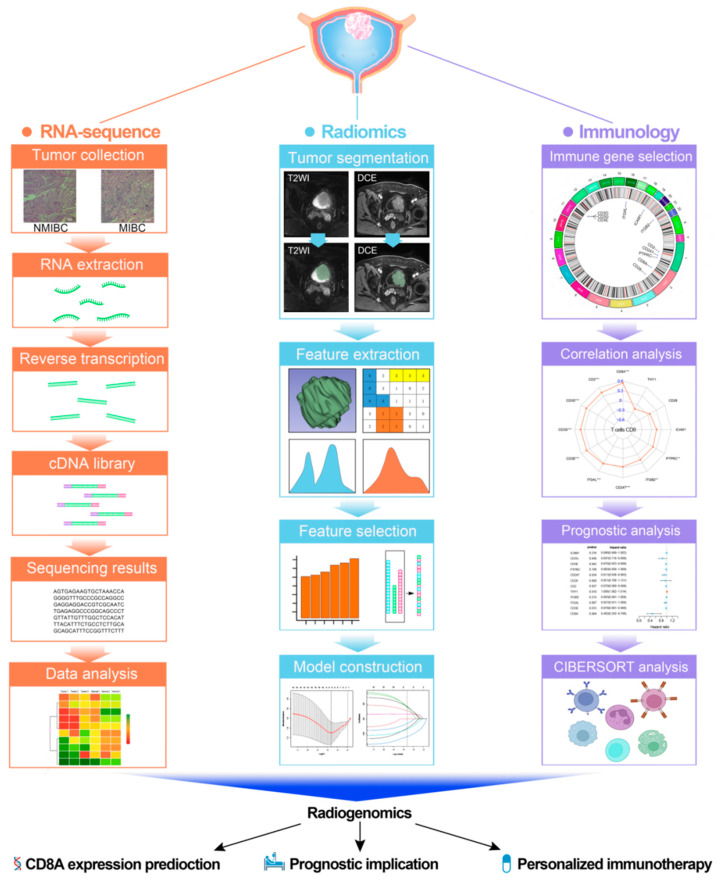
The work flow of this study. NMIBC: non-muscle-invasive bladder carcinoma; MIBC: muscle-invasive bladder carcinoma.

**Figure 2 cancers-14-04866-f002:**
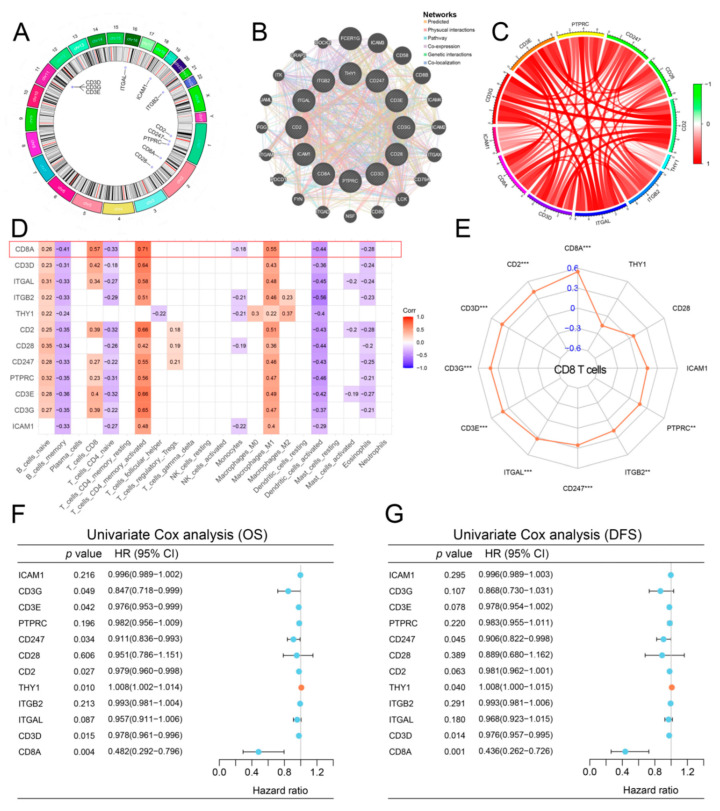
Identification of hub gene of T cytotoxic pathway in BCa. (**A**) Locations of T cytotoxic pathway-related genes on 23 chromosomes. (**B**) Interaction of T cytotoxic pathway-related genes. (**C**) Correlation analysis among T cytotoxic pathway-related genes. (**D**) Correlation analysis between T cytotoxic pathway-related genes and TIICs. (**E**) The association of the CD8+ T cells with T cytotoxic pathway-related genes. (**F**,**G**) Univariate Cox analyses of the T cytotoxic pathway-related genes in BCa. BCa, bladder cancer; TIICs, tumor-infiltrating immune cells; OS, overall survival; DFS, disease-free survival. **: *p* < 0.01, ***: *p* < 0.001.

**Figure 3 cancers-14-04866-f003:**
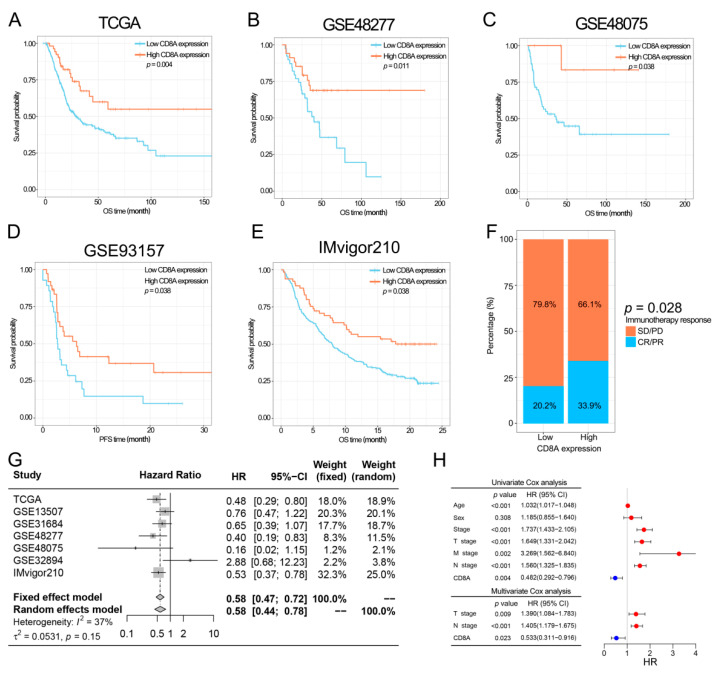
Investigating the clinical predictive value of CD8A in prognosis and immunotherapeutic response. (**A**–**E**) Kaplan–Meier plots of CD8A in five datasets, including TCGA (**A**), GSE48277 (**B**), GSE48075 (**C**), GSE93157 (**D**) and IMvigor210 (**E**). (**F**) Percentages of the different PD-L1 inhibitor responses between high or low CD8A expression groups in IMvigor210. (**G**) Forest plot of the HRs for patients with high CD8A expression compared to patients with low CD8A expression in seven BCa datasets. (**H**) Univariate and multivariate Cox analyses in TCGA-BCa. OS, overall survival; PFS, progression-free survival; CR, complete response; PR, partial response; SD, stable disease; PD, progressive disease; HR: hazard ratio; TCGA: The Cancer Genome Atlas; BCa, bladder cancer.

**Figure 4 cancers-14-04866-f004:**
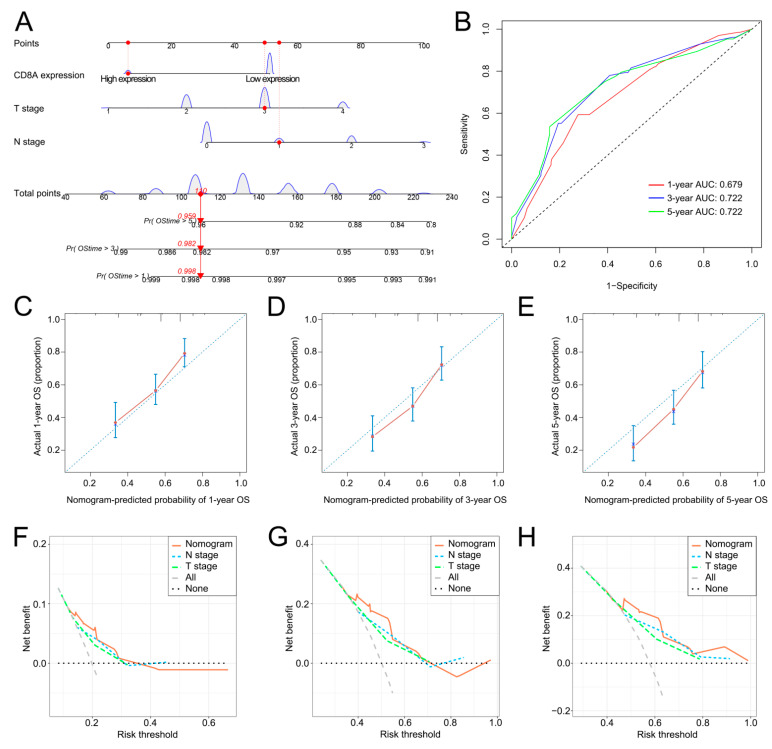
The construction and performance of the CD8A-based nomogram in TCGA. (**A**) CD8A-based nomogram for predicting the probability of 1-, 3- and 5-year OS. (**B**) ROC curves to predict the 1-, 3- and 5-year OS according to the CD8A-based nomogram. (**C**–**E**) Calibration plots of the CD8A-based nomogram for predicting the probability of 1-, 3- and 5-year OS. (**F**–**H**) DCA of the CD8A-based nomogram predicting 1-, 3- and 5-year OS. TCGA, The Cancer Genome Atlas; OS, overall survival; BCa, bladder cancer; ROC, receiver operating characteristic curve; DCA, decision curve analysis.

**Figure 5 cancers-14-04866-f005:**
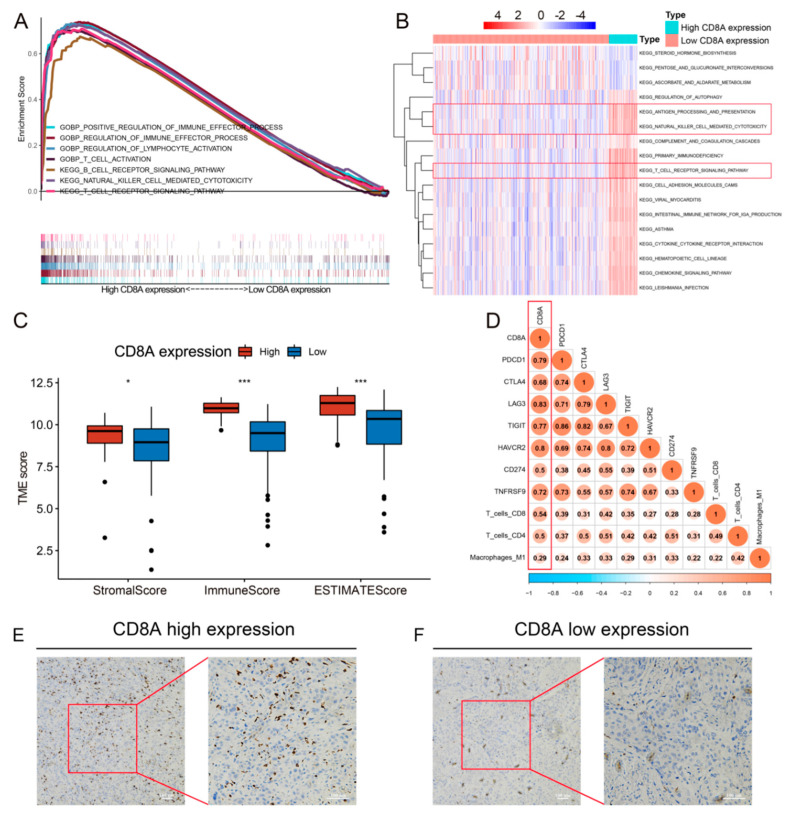
Biological analysis of CD8A. (**A**) Gene set enrichment analysis for comparing biological pathways between high and low CD8A expression groups. (**B**) Gene set variation analysis analyzed the different biological pathways between high and low CD8A expression groups. (**C**) The different levels of TME scores between low and high CD8A expression groups. (**D**) The association of CD8A with TIICs and critical immune checkpoint genes. (**E**) Patient 1: an 88-year-old man with MIBC and IHC-based CD8A high expression. (**F**) Patient 2: a 65-year-old man with MIBC and IHC-based CD8A low expression. TME, tumor microenvironment; TIICs, tumor-infiltrating immune cells; IHC, immunohistochemistry; MIBC, muscle-invasive bladder carcinoma. *: *p* < 0.05, ***: *p* < 0.001.

**Figure 6 cancers-14-04866-f006:**
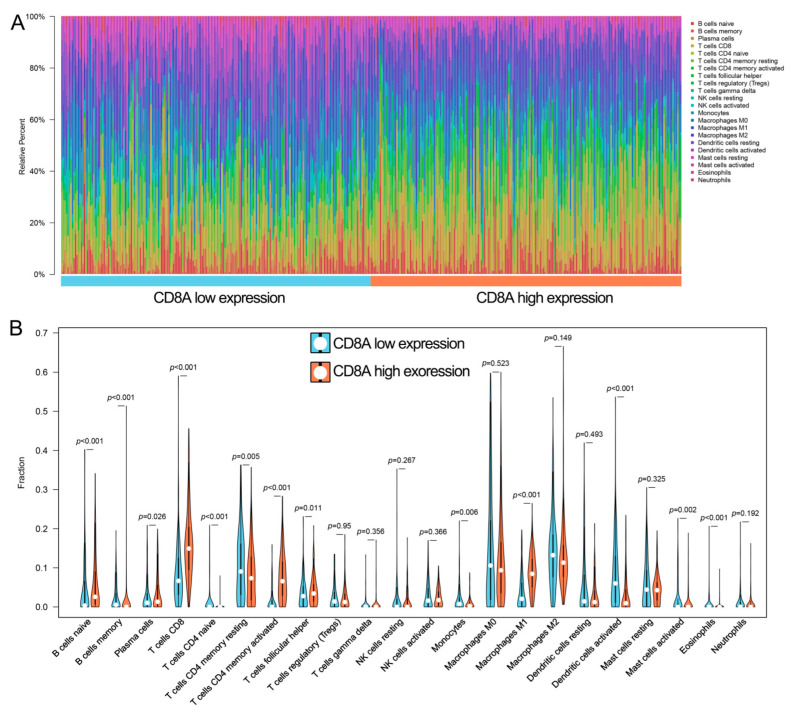
The landscape of 22 TIICs in TCGA-BCa. (**A**) The proportions of 22 TIICs in each sample quantified by CIBERSORT algorithm. (**B**) The difference of the proportions of 22 TIICs between high and low CD8A expression groups. TCGA, The Cancer Genome Atlas; BCa, bladder cancer; TIICs, tumor-infiltrating immune cells; CIBERSORT, cell type identification by estimating relative subsets of RNA transcripts.

**Figure 7 cancers-14-04866-f007:**
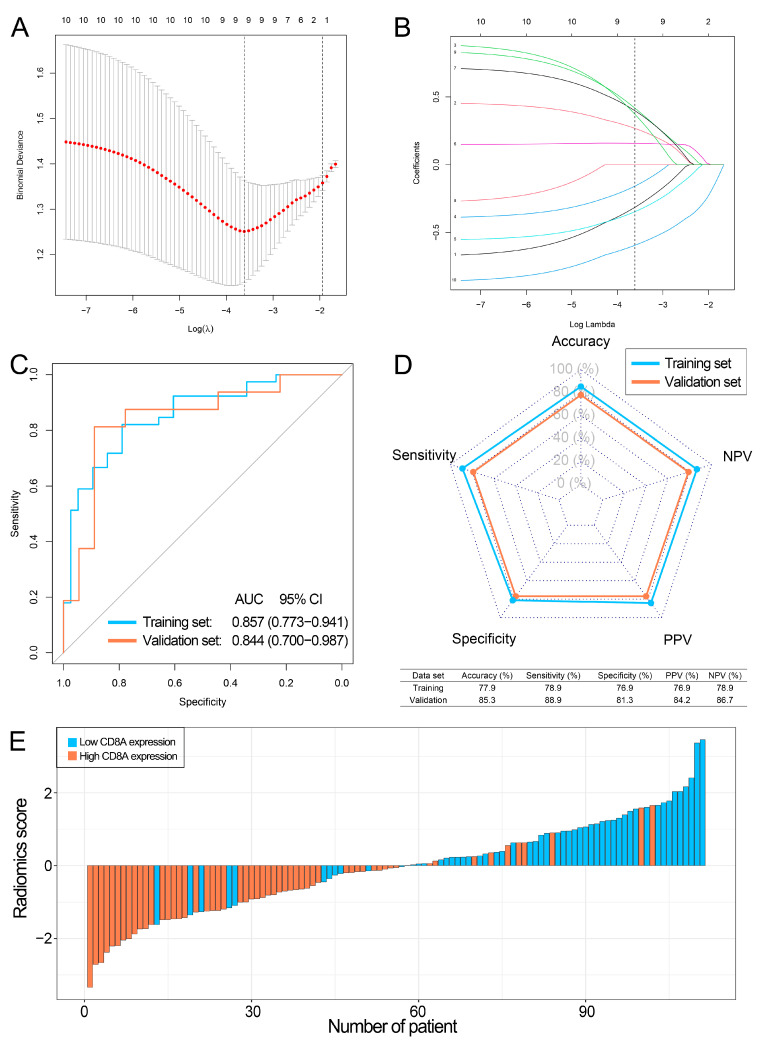
Construction and performance of the LASSO model (radiomics signature) in our center. (**A**) Optimal radiomics features selection based on the binominal deviance. (**B**) Distribution of LASSO coefficients for nine selected radiomics features. (**C**) AUCs of the LASSO model in the training and validation sets. (**D**) The comprehensive performance of the LASSO model in the training and validation sets. (**E**) Waterfall plot of the CD8A expression levels and radiomics scores in the combined training and validation sets. LASSO, least absolute shrinkage and selection operator; NPV, negative predictive value; PPV, positive predict value; AUCs, area under the ROC curves.

**Figure 8 cancers-14-04866-f008:**
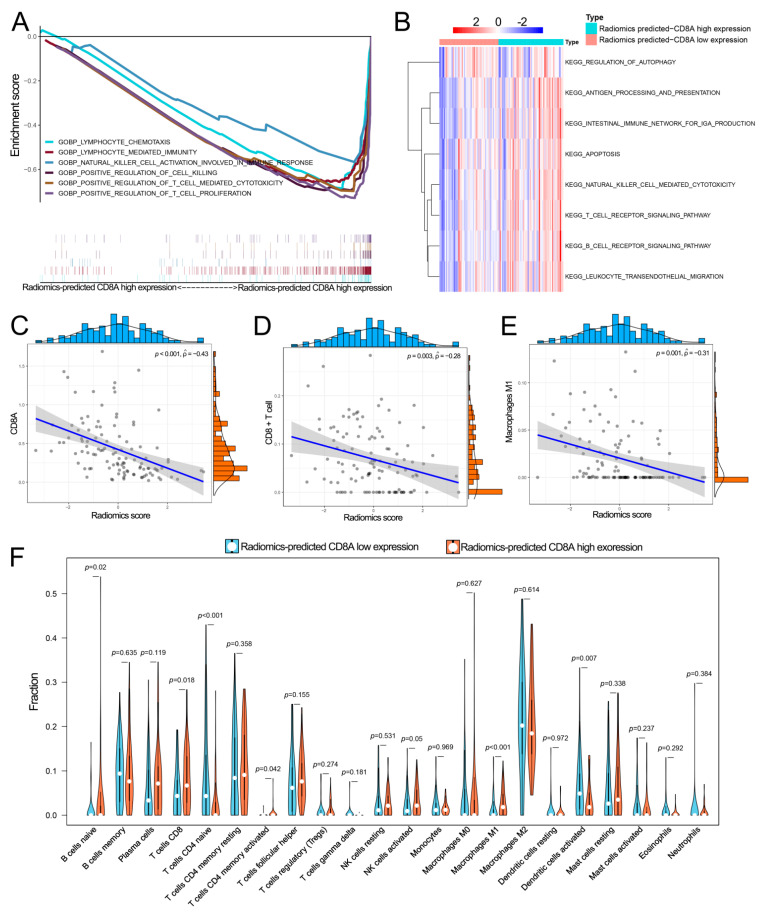
Biological analysis of the radiomics signature (LASSO model). (**A**,**B**) Gene set enrichment analysis (**A**) and gene set variation analysis (**B**) for comparing biological pathways between radiomics-predicted high and low CD8A expression groups. (**C**–**E**) The association of radiomics score with CD8A expression, CD8A (**C**) and degree of infiltration of CD8+ T cells (**D**) and Macrophages M1 (**E**). (**F**) The difference of the proportions of 22 TIICs between radiomics-predicted high and low CD8A expression groups. LASSO, least absolute shrinkage and selection operator; TIICs, tumor-infiltrating immune cells.

**Table 1 cancers-14-04866-t001:** Clinicopathological characteristics of patients in our center.

	Number of Patients (%)		
Characteristic	Training Set (n = 77)	Validation Set (n = 34)	*p* Value ^b^
Sex			
Men	62 (80.5)	29 (85.3)	0.546
Women	15 (19.5)	5 (14.7)	
Age (years)			
<65	23 (29.9)	10 (29.4)	0.961
≥65	54 (70.1)	24 (70.6)	
Tumor size ^a^ (cm)			
<3	38 (49.4)	17 (50)	0.950
≥3	39 (50.6)	17 (50)	
Number of tumors ^a^			
Single	52 (67.5)	23 (67.6)	0.991
Multiple	25 (32.5)	11 (32.4)	
Pathological grade			
Low grade	15 (19.5)	9 (26.5)	0.410
High grade	62 (80.5)	25 (73.5)	
Clinical T stage			
<T2	50 (64.9)	23 (67.6)	0.781
≥T2	27 (35.1)	11 (32.4)	

^a^ MRI-determined information. ^b^
*p* value from chi-square test.

## Data Availability

The processed data required to reproduce these findings cannot be shared at this time as the data also form part of an ongoing study. Requests to access the datasets should be directed to Shenghua Liu, drfelixliu@163.com.

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
