# Peer review of "CD8A as a Prognostic and Immunotherapy Predictive Biomarker Can Be Evaluated by MRI Radiomics Features in Bladder Cancer"

_cancers, 2022, doi:10.3390/cancers14194866_

Round 1

Reviewer 1 Report

This is a well-written manuscript that has used publically available data sets and then deconvoluted and amalgamated the sequencing information with radiomics and immunology. Although CD8 has been well known as an important biomarker in TME in bladder cancer and CD8a has been described recently in context to radiomics in many cancers. This manuscript has tried bringing this angle into the field of bladder cancer. Most of the figures and the results are well displayed with the overall conclusion of the study well laid out. I would have additionally liked to see validation of these observations via a simple Immunohistochemistry (CD8a IHC) on prototypical representative cases versus a CD8a negative case. This will bring a good extra validation to the information presented in the manuscript. 

Also, the addition of simple H&E microphotographs as MIBC versus NMIBC  would have been a nice touch and generated and explained to the potential readers regarding the disease process.    

Author Response

Thank you for your comments. We have performed the immunohistochemistry of CD8A on bladder cancer tissues. The representative immunostaining for low and high CD8A expressions was shown in Figure 5E,F. In addition, the H&E microphotographs of NMIBC and MIBC were shown in Figure 1.

We appreciate your warm work earnestly, your comments are valuable and helpful for improving our paper, as well as the important guiding significance to readers. We hope that the correction will meet with approval.

Reviewer 2 Report

Thank you for inviting me to review the article titled “ CD8A as a prognostic and immunotherapy biomarker can be evaluated by MRI radiomics features in bladder cancer

The authors are developing a new radiogenomics signature to predict CD8A expression and assess predictive and prognostic values of CD8A expression

Criticisms:

Major:  

Paper is trying to do too much.

9 figures with all of them having sub figures (~ 30 figures) while important is a lot for reader to focus on. Authors should seriously consider breaking the paper into 2 papers where paper 1 establishes CD8A expression as a marker and then use a separate paper to create radiogenomics for bladder ca. As in the current paper information about bladder cancer and radiogenomics feels like an afterthought.

Minor:

Abstract lacks structure and information

Line 72: Unsure of the need of COVID-19 and CD8A reference, there is no need for it

Line 252 – IC50 needs full form

Line 272 – sentence needs a new structure

Line 473 – correction: Zeng et al

Author Response

Thank you for your valuable comment.

We admitted that there were quite a lot of figures and sub-figures for readers to understand and focus on. To solve this problem, we put some sub-figures in supplementary files and merge some sub-figures into a more comprehensive sub-figure. These changes will not influence the content and framework of the paper.

As for breaking the paper into 2 papers, we studied this comment carefully and summarized three reasons for submitting a comprehensive paper instead of breaking the paper into 2 papers:

  1. As shown in Figure 1, Radiogenomics contains genomics analysis (including RNA-sequence and immunology in this study) and radiomics analysis. In this study, we first found that CD8A is a novel indicator for predicting the prognosis and immunotherapy biomarker in bladder cancer through genomics analysis, which reveals the clinical application value of this gene. In order to apply these results in clinical practice, we then constructed the MRI-based radiomics signature to preoperatively predict the expression of CD8A in bladder cancer patients, thereby contributing to the prediction of prognosis and immunotherapeutic susceptibility. Only in this way, a comprehensive radiogenomics analysis was performed.
  2. There were only two radiogenomics-related Figures (Figure 7 and Figure 8) in the revised manuscript. If we break the paper into 2 papers, the contents of one paper containing the radiogenomics would not be abundant.
  3. There were many articles (such as: PMID: 33761371, PMID: 33416947 and PMID: 33811161) that have the same framework as ours, which could prove the rationality of the content and framework in our manuscript.

As for the minor comments:

We have revised the Abstract section to present the information of the study in a more organized way.

In Line 72, we have removed this sentence and the corresponding references.

Line 252, we have replaced the ‘IC50’ with ‘half maximal inhibitory concentration (IC50)’.

In Line 272, we have replaced this sentence with a more structured sentence.

Line 473, we have replaced the ‘Zeng’ with ‘Zeng et al’.

Reviewer 3 Report

It’s a well written paper on novel concept, with appropriate methodology. Some minor comments were listed below:

In Line 76 The latest study showed that CD8A is involved in coronavirus biology and is rele- 76 vant for COVID-19 prognosis[15, 16]. The sentence is not connected with the study topic and does not contain any relevant information. Yet, is there any connections with transitional cell carcinoma and CD8a expression studied?

In Lines 83-85: what about developed protocol of VI-RADS (e.g. 10.2214/AJR.20.22763)? could you comment it in the discussion? What’s the advantage of your method over VI-RADS? What’s the cost-effectiveness of your method when combined with VI-RADS as the latter does not need any additional tools than contrast and adequate protocol?

Author Response

Thank you for your comments. We have removed the sentence and the corresponding references in Line 76. Through the literature review, we found that there are no articles investigating the connections between transitional cell carcinoma and CD8a expression.

As for the VI-RADS, it is developed to provide accurate information for the diagnosis of muscle-invasive bladder cancers (MIBCs) based on multiparametric magnetic resonance imaging (mpMRI). In our previous articles, we have validated that VI-RADS is a promising and effective modality in determining detrusor muscle invasion of bladder cancer preoperatively (PMID: 32420150), and combined mpMRI-based radiomics signature with the VI-RADS to preoperatively differentiate muscle invasion of bladder cancer (PMID: 34268106).

However, VI-RADS is an imaging characteristic generated by radiologists’ subjective evaluation, and it is mainly used for the diagnosis of MIBC. On the contrary, radiomics is more objective and can extract high-dimensional radiomics features that could not be detected by human eyes and might be correlated with tumor biology. In our previous article (PMID: 34863282), we develop and validate MRI-based radiomics signatures to preoperatively predict the Ki67 expression status in bladder cancer, which revealed the possibility of radiomics in biological behaviors prediction.

In this study, the MRI-based radiomics signatures achieved the AUCs of 0.857 and 0.844 in the training and validation sets, respectively. MRI-based radiomics signatures have the potential to preoperatively predict the expression of CD8A in bladder cancer patients.

The above information about the VI-RADS and the reference (e.g. 10.2214/AJR.20.22763) have been added to the Introduction and Discussion sections. We appreciate your warm work earnestly. We hope that the correction will meet with approval.

Round 2

Reviewer 2 Report

Thank you for all the hard work and for addressing the comments.

While the paper is a tough read mainly because it is long, but does establish a point. Thank you for trying to concise the manuscript further.

If any sentence refers to the assessment of CD8A expression as a predictor of immunotherapy response, then it is considered a "predictive" marker, and it should be used in that context.

Line 542:  Sentence needs grammatical correction. 

Line 629: Sentence needs grammatical correction. 

Author Response

Thank you for your comment. We have replaced the word ‘biomarker’ with ‘predictive marker’ in any sentence refers to the assessment of CD8A expression as a predictor of immunotherapy response.

We are sorry for our grammatical mistakes in Line 542 and Line 629. We have made grammatical corrections in Line 542 and Line 629. In addition, we also comprehensively reviewed the manuscript again and made several grammatical corrections marked by ‘Track Changes’ in the revised manuscript.

We appreciate for your warm work earnestly, your comments improves the readability and intelligibility of our manuscript. We hope that the correction will meet with approval.